# Risk assessment of atherosclerotic cardiovascular diseases before statin therapy initiation: Knowledge, attitude, and practice of physicians in Yemen

**Fahmi Y. Al-Ashwal**[1,2]\*, **Syed Azhar Syed Sulaiman**[1]\*, **Siti Maisharah Sheikh Ghadzi**[1], **Mohammed Abdullah Kubas**[2], **Abdulsalam Halboup**[3]

1 Discipline of Clinical Pharmacy, School of Pharmaceutical Sciences, Universiti Sains Malaysia, Penang, Malaysia, 2 Clinical Pharmacy Department, University of Science and Technology Hospital (USTH), Sana'a, Yemen, 3 Department of Clinical Pharmacy and Pharmacy Practice, Faculty of Pharmacy, University of Science and Technology, Sana'a, Yemen

\* fahmialashwal89@gmail.com (FYAA); sazhar@usm.my (SASS)

## Abstract

### Background

Risk evaluation of atherosclerotic cardiovascular disease (ASCVD) remains the cornerstone of primary prevention. The cardiovascular risk assessment can guide the decision-making on various preventive measures such as initiating or deferring statin therapy. Thus, our study aimed to assess the physicians' knowledge, attitude, and practices regarding atherosclerotic cardiovascular diseases risk assessment. Also, we evaluated the physician-patient discussion and counseling practices before statin therapy initiation in concordance with recommendations from the latest clinical practice guideline.

### Methods

A cross-sectional study was conducted between November 2020 and January 2021. A self-administered questionnaire was distributed to 350 physicians (GPs, residents, specialists, and consultants). Two trained pharmacists distributed the questionnaires in 5 major tertiary governmental hospitals and more than ten private hospitals. Also, private clinics were targeted so that we get a representative sample of physicians at different workplaces.

### Results

A total of 270 physicians filled the questionnaire out of 350 physicians approached, with 14 being excluded due to high missing data, giving a final response rate of 73%. Participants had suboptimal knowledge and practices with a high positive attitude toward atherosclerotic cardiovascular diseases risk assessment. The knowledge and practices were higher among consultants, participants from the cardiology department, those with experience years of more than nine years, and those who reported following a specific guideline for cholesterol management or using a risk calculator in their practice. Notably, the risk assessment and

**Data Availability Statement:** All relevant data are within the paper and its Supporting information files.

**Funding:** The study was funded by University of Sciences and Technology, Sana'a, Yemen. The funders had no role in study design, data collection and analysis, decision to publish, or preparation of the manuscript.

**Competing interests:** The authors have declared that no competing interests exist.

counseling practices were lower among physicians who reported seeing more patients per day.

## Conclusion

Physicians had overall low knowledge, suboptimal practices, and a high positive attitude toward cardiovascular risk assessment. Therefore, physicians' training and continuing medical education regarding cholesterol management and primary prevention clinical practice guidelines are recommended. Also, the importance of adherence to clinical practice guidelines and their impact on clinical outcomes should be emphasized.

## Introduction

Cardiovascular disease (CVD) is a highly prevalent condition and a major contributor to health loss. CVD remains the leading cause of global mortality, representing more than 30% of global deaths in 2015 [1]. According to the World Health Organization (WHO), an estimated 17.9 million people died in 2016 from CVDs, representing 31% of all deaths worldwide. Notably, 85% percent of these deaths are due to heart attack and stroke, and over three-quarters of CVD deaths occur in low- and middle-income countries [2]. The Institute for Health Metrics and Evaluation (IHME) has shown that the top leading cause of death in the Arab world is CVDs [3]. Also, risk factors for CVD, such as obesity and diabetes mellitus, are common, and they have been on growth throughout the world [4]. Noticeably, CVD exacts a heavy burden not only on the patients but also on their families and the governments [5–8]. Accordingly, prevention and reversing the growth of CVD is a public health priority.

Risk evaluation of atherosclerotic cardiovascular disease (ASCVD) remains the cornerstone of primary prevention. The current clinical practice guidelines on the management of dyslipidemia and primary prevention of CVD recommend a risk assessment of CVD for eligible patients [9, 10]. The CV risk can be assessed using risk estimation algorithms created based on the results of cohort studies [11]. Different risk score calculators are recommended by different guidelines for assessing the 10-year cardiovascular risk [9, 12]. These risk calculators differ in the variables included and the endpoints assessed [11, 13]. For example, the 2008 Framingham General CVD risk calculator uses the variables of gender, age, total cholesterol, HDL cholesterol, systolic blood pressure, antihypertensive therapy, history of diabetes mellitus, and current smoking status [11, 13]. The outcomes being assessed are the total CVD (coronary insufficiency or angina, heart failure, Intermittent claudication, CHD death, nonfatal MI, fatal or nonfatal ischemic or hemorrhagic stroke, and transient ischemic attack). The 2013 ACC/AHA risk calculator includes almost the same parameters as the 2008 Framingham general CVD model, but in contrast to the 2008 Framingham model, it adds the race and measures only hard ASCVD endpoints (CHD death, nonfatal MI, fatal and nonfatal stroke) [11, 13].

The 10-year cardiovascular risk assessment help to guide decision-making on various preventive measures such as initiating or deferring statin therapy. Also, calculating the 10-year ASCVD risk of a patient enables the healthcare providers to adjust the intensity of preventive measures to the patients' risk. In this light, the 2018 AHA/ACC guideline on the management of dyslipidemia recommends that a 10-year risk calculation should be performed for adult patients aged 40–75 years old who are free of ASCVD. Also, it advocates for a lifetime risk calculation for younger individuals [9]. For patients with DM, ASCVD, and primary

hypercholesterolemia, risk assessment is not needed but can be used to intensify statin therapy in patients with diabetes mellitus (DM) [10].

Physicians play essential roles in the prevention and management of CVD. Therefore, having adequate knowledge and positive attitudes towards CV risk assessment are of vital importance for their practice to improve patients outcomes. Few previous studies from America, Singapore, and Jordan evaluated the physicians' knowledge and attitudes regarding the 2013 ACC/AHA cholesterol guideline [14–16]. However, data regarding their knowledge, attitude, and practice towards CVD risk assessment before initiating statin therapy are scarce, especially in the Middle East. Therefore, this study aimed to evaluate the knowledge, attitude, and practices of Yemeni physicians regarding risk assessment of atherosclerotic cardiovascular diseases before initiating statin therapy.

## Methods

### Ethical approval

Ethical approval for this study was granted by the Ethical Committee of the Medical Research, University of Sciences and Technology, Sana'a, Yemen (EAC/UST193). The ethical committee approved verbal informed consent, and participants who consented were included in the study. Study objectives were explained adequately to all participants.

### Study design and setting

A cross-sectional study was conducted using a structured validated questionnaire between November 2020 and January 2021. The study was done in the capital of Yemen, Sana'a. To approach physicians, two trained pharmacists distributed the questionnaires in 5 major tertiary governmental hospitals and twelve private hospitals. Also, private clinics were targeted so that we get a representative sample of physicians at different workplaces.

### Sample size calculation and participants

The current study's target population consisted of 1732 physicians, according to the last annual health report of the number of physicians in Sana'a [17]. The total sample size was calculated to be 214 based on the following formula $N = 4 Z_\alpha^2 S^2 \div W^2$ [18], assuming a 95% confidence interval, $Z_\alpha$ of 1.96, W/S ratio of 0.3 [15], and 20% for non-responses or in case of incomplete questionnaires. A total of 350 questionnaires were distributed. The study was carried out among physicians most likely to be involved in statin prescription. These include providers from internal medicine, cardiology, endocrinology, nephrology departments, and general practitioners. The targeted physicians were categorized into consultants (those who have a subspecialty), specialists (physicians who completed four or five years of residency program), residents (physicians enrolled in a 4 or 5-year residency program), and general practitioners (licensed physicians who are graduated from an accredited medical school without being enrolled into a residency program).

### Data collection tool

A self-administered questionnaire was designed based on information and recommendations for ASCVD risk assessment and statin therapy initiation according to the latest guidelines. These include the 2018 AHA/ACC Guideline on the Management of Blood Cholesterol and the 2019 AHA/ACC Guideline on the Primary Prevention of Cardiovascular Disease [9–11]. Also, a few relevant questions were adapted from previous literature [19].

The questionnaire consists of 6 sections (S1 File). Section A contained data about gender, age, working place, specialty, and experience years. Moreover, four general questions were included as follows: 'Number of patients seen per day?', 'In the past month, how many times did you prescribe statin therapy?', 'Do you follow any clinical practice guideline for cholesterol management in your patients?', and 'Do you use a risk calculator for cardiovascular risk assessment in your practice?'. Section B contained 6 questions that assessed the general awareness about the 2018 ACC/AHA guideline, Framingham general CVD risk calculator, and the 10-year ASCVD risk calculator.

Section C assessed the specific knowledge regarding ASCVD risk assessment. It included 10 multiple-choice questions that were designed to assess whether physicians have the basic and necessary knowledge for risk assessment before statin therapy initiation. Section D included 7 questions in which participants were asked about their attitude towards CVD risk assessment, with responses being measured on a 5-point Likert scale: strongly disagree, disagree, neutral, agree, and strongly agree. Section E contained 3 questions that evaluated the practices for risk assessment in Yemen with five possible responses (never, rarely, sometimes, often, always). The last section (F) included 10 questions that evaluated the counseling practices of physicians before statin therapy initiation (patient-physician discussion), with responses being measured on a 5-point Likert scale: never, rarely, sometimes, often, always.

## Scoring system

For the knowledge section, the correct answer was coded as 1, and the wrong or 'I do not know' answer was coded as 0. The total score ranges from zero to ten. For the attitude, the seven questions on a 5-point Likert scale were coded into 1–5, from strongly disagree to strongly agree, respectively. Accordingly, the total scores ranged from 7 to 35. For the counseling practices (patient-physician discussion), the 10 questions were scored into 1–5, from never to always, respectively; and the total scores ranged from 10 to 50.

## Validation

The questionnaire was given to 6 experts in clinical pharmacy, community medicine, pharmacy practice, and internal medicine (3 consultants) for content validation. The experts were asked to assess the relevance and representative of each item to its domain. The Scale-Content Validity Index based on the Universal Agreement method (S-CVI/UA) for general awareness, knowledge, attitude, and counseling practices domains were 1, 0.80, 1, and 1, respectively. This indicates a satisfactory level of content validity for the domains [20]. For face validation, three physicians and four pharmacists assessed the clarity and comprehension of the questions in each domain. Then, the questionnaire was piloted-tested on 34 physicians to assess its reliability. The calculated Cronbach's alpha for the awareness, attitude, risk assessment practices, and patients-physician discussion practice were 0.70, 0.81, 0.75, and 0.71, respectively.

## Statistical analysis

Data were analyzed using SPSS, version 25.0 (IBM Corp., Armonk, NY, USA). Both inferential and descriptive analyses were utilized for this study. Frequency (percentages) was used for categorical variables, and median (interquartile range) was utilized for the overall scores. To assess the association between participants' demographic data and their overall knowledge, attitude, and practices, we used the Mann-Whitney $U$ test and the Kruskal-Wallis test as appropriate. A $P$ value below 0.05 indicated a statistically significant difference.

## Results

A total of 350 physicians were approached, and 270 filled out the questionnaire, with 14 being excluded due to high missing data, giving a final response rate of 73%. Table 1 displays the demographic characteristics of the participants. Approximately 64% of participants were

**Table 1. Demographics characteristics (n = 256).**

| Parameter | Frequency (%) |
|---|---|
| **Gender** | |
| Male | 164 (64.1) |
| Female | 92 (35.9) |
| **Age (Years)** | |
| < 37 | 119 (46.5) |
| ≥ 37 | 125 (48.8) |
| Missing | 12 (4.7) |
| **Experience (Years)** | |
| <9 | 125 (48.8) |
| ≥9 | 123 (48) |
| Missing | 8 (3.1) |
| **Current position** | |
| Consultant | 49 (19.1) |
| Specialist | 59 (23.1) |
| Resident doctor | 50 (19.5) |
| General practitioner | 98 (38.3) |
| **Department** | |
| Cardiology | 56 (21.9) |
| Nephrology | 23 (9) |
| Internal medicine | 110 (42.9) |
| Others | 67 (26.2) |
| **Current workplace** | |
| Private hospital | 82 (32) |
| Governmental hospital | 118 (46.1) |
| Private clinic | 56 (21.9) |
| **Number of patients seen per day** | |
| ≤ 25 | 90 (35.2) |
| > 25 | 166 (64.8) |
| **The number of statin therapy prescriptions in the past month?** | |
| ≤ 25 | 107 (41.8) |
| > 25 | 149 (58.2) |
| **Do you follow any clinical practice guideline for cholesterol management in your patients?** | |
| Yes | 137 (53.5) |
| No | 119 (46.5) |
| **For those who answered 'Yes', the guidelines they usually follow:** | |
| The American College of Cardiology/American Heart Association (AHA/ACC) Guideline on the management of blood cholesterol | 123 (89.8) |
| The European Society of Cardiology/European Atherosclerosis Society (ESC/EAS) guideline | 11 (8) |
| The National Institute for Health and Care Excellence (NICE) guideline | 3 (2.2) |
| **Do you use a risk calculator for cardiovascular risk assessment in your practice?** | |
| Yes | 84 (32.8) |
| No | 172 (67.2) |

males, and almost half of them (48.8%) had less than 9 years of experience. More than two-fifths of physicians (46.1%) were from governmental hospitals, and nearly a third of them worked in private hospitals (32%). Over 40% of participants were from the internal medicine department, and only 9% were from the nephrology department. The respondents were mainly general practitioners (38.3%), followed by specialists (23.1%), residents (19.5%), and consultants (19.1%). Almost half of the respondents were ≥37 years old. A considerable percentage of physicians (64.8%) reported they see more than 25 patients a day. Almost three-fifths of respondents (58.2%) had more than 25 statin therapy prescriptions in the past month prior to data collection. Notably, only 53.5% of participants were following a specific guideline for their patients' cholesterol management, the majority of which (89.8%) were following the ACC/AHA guideline on the management of blood cholesterol. Surprisingly, a huge percentage of participants (67.2%) did not use any risk calculator for cardiovascular risk assessment in their practice.

## General awareness of the guideline and risk calculators

Only 43.8% of physicians stated that they read either the summary or the full report of the 2018 ACC/AHA guidelines on the management of blood cholesterol. Just over a third of participants (34.8%) knew of any differences between the 2018 and 2013 ACC/AHA guidelines for cholesterol management (Table 2). Similarly, 42.2% of participants were aware of the parameters used in the Framingham CVD risk calculator. Moreover, less than a third (30.5%) were familiar with the parameters used by the ASCVD risk calculator. Of note, 26.6% of respondents stated they know the differences between the two risk calculators. Furthermore, only two-fifths of physicians (40.6%) were aware of the web version or the downloadable ASCVD 10-year risk calculator.

## Knowledge about ASCVD risk assessment

Participants had an average level of knowledge with an overall median (IQR) score of 5 (4–6) out of 10. The majority of participants (71.1%) knew the age category for which a 10-years risk assessment is recommended. However, only a third of physicians (33.6%) identified the age category for which a lifetime risk assessment is advocated instead of a 10-year risk calculation. A significant proportion of participants (70.3%) correctly identified the risk category (very high risk) for a 65-year-old smoker patient with a history of myocardial infarction. In contrast, only half of them (49.6%) were able to identify the risk category for a 40-year-old diabetic patient. Other knowledge gaps have been identified among the physicians (Table 3). In this light, nearly two-fifths of participants (39.5%) did not know that chronic inflammatory

**Table 2. Response of physicians to the general awareness questions about guidelines and risk calculators.**

| Statement | Frequency (%) |
|---|---|
| Physicians have read either the summary or the full report of the 2018 ACC/AHA guidelines on the management of blood cholesterol: | 112 (43.8) |
| Physicians were aware of any differences between the 2018 ACC/AHA guideline and the 2013 ACC/AHA guideline on the management of blood cholesterol | 89 (34.8) |
| Physicians were aware of the parameters used in the Framingham CVD risk calculator | 108 (42.2) |
| Physicians were aware of the parameters used in the ACC/AHA 10-year ASCVD risk calculator (Pooled Cohort Equations (PCE)) | 78 (30.5) |
| Physicians were aware of any differences between the Framingham General CVD risk calculator and the ACC/AHA ASCVD 10-year risk calculator | 68 (26.6) |
| Physicians aware of the web version or the downloadable ASCVD 10-year risk calculator | 104 (40.6) |

**Table 3. Physicians' knowledge about ASCVD risk assessment before statin therapy initiation.**

| | Frequency (%) | Overall score Median (IQR) |
|---|---|---|
| Physicians knew the age category for which a 10-year risk calculation is recommended for primary prevention of ASCVD | 182 (71.1) | 5 (4–6) |
| Physicians knew the age category for which a lifetime risk assessment is recommended instead of a 10-year risk calculation | 86 (33.6) | |
| Physicians were able to identify the 4 categories for risk stratifications according to the 2018 AHA/ACC guidelines | 88 (34.4) | |
| Physicians able to identify the risk category for a 40-year-old diabetic patient | 127 (49.6) | |
| Physicians were able to identify the risk category for a 65-year-old smoker patient with a history of myocardial infarction | 180 (70.3) | |
| Physicians know that chronic inflammatory conditions enhance the individual ASCVD risk | 155 (60.5) | |
| Physicians knew that the AHA/ACC 10-year risk calculator may underestimate risk in patients with chronic inflammatory conditions | 125 (48.8) | |
| Physicians were aware that for individuals with intermediate-risk, the coronary artery calcium (CAC) score can be useful to refine the risk and aid in decision making about statin initiation | 68 (26.6) | |
| Physicians knew that a non-fasting plasma lipid profile is effective in estimating ASCVD risk in adults not on lipid-lowering therapy | 144 (56.3) | |
| Physicians were able to identify the 4 outcomes captured by the AHA/ACC 10-year risk calculator (pooled cohort equations) | 44 (17.2) | |

conditions are considered risk enhancers according to the latest AHA/ACC clinical practice guideline. In addition, over half of the participants (51.2%) were not aware that the ASCVD risk calculator underestimates the cardiovascular risk in patients with chronic inflammatory conditions. Moreover, over two-fifths of respondents (43.7%) were not aware of the effectiveness of non-fasting plasma profile in estimating ASCVD risk in individuals not on lipid therapy. Furthermore, only a minority (17.2%) of participants identified the four outcomes captured by the AHA/ACC 10-year risk calculator, and just over a quarter (26.6%) identified the risk category (intermediate risk) for which a coronary artery calcium (CAC) score is helpful to refine the risk and aid in decision making about statin therapy.

A Mann-Whitney U test (Table 4) revealed that the knowledge score was significantly higher among participants with experience years $\geq 9$ ($U = 6387$, p = 0.018), those following a specific guideline for cholesterol management ($U = 5858$, p = 0.001), and those who were using a risk calculator in their clinical practice ($U = 5384$, p = 0.001). The Kruskal-Wallis test (Table 5) showed a significant difference ($\chi^2 = 26.921$, df = 3, p = 0.001) among the position groups with a mean rank knowledge score of 174.69 for consultants, 121.31 for specialists, 126 for residents, and 110.87 for GPs. Dunn's pairwise tests were carried out and revealed that consultants had a significantly higher knowledge score than specialists, residents, and GPs. Moreover, a significant difference in knowledge ($\chi^2 = 23.893$, df = 3, p = 0.001) was associated with departments with a mean rank knowledge score of 169.2 for cardiology, and 115.41 for nephrology, 121.51 for internal medicine, and 110.46 for others. Dunn's pairwise tests revealed that physicians from the cardiology department had significantly higher knowledge scores than those from nephrology, internal medicine, and others.

## Attitude towards ASCVD risk assessment

Generally, participants had a positive attitude towards ASCVD risk assessment with an overall median (IQR) score of 28 (27–30) out of 35 (Table 6). In this light, the majority of participants

**Table 4. The data of the Mann-Whitney *U* test for the association between demographic variables (with two categories) and physicians' knowledge, attitude, and practices.**

| Variable | | Knowledge | | | | Attitude | | | | Practice (Risk assessment) | | | | Practice (Patient-physician counseling) | | | |
|---|---|---|---|---|---|---|---|---|---|---|---|---|---|---|---|---|---|
| | N | Mean rank | Test value (U) | z | p-value | Mean rank | Test value (U) | z | p-value | Mean rank | Test value (U) | z | p-value | Mean rank | Test value (U) | z | p-value |
| **Gender** | | | | | | | | | | | | | | | | | |
| Male | 164 | 132.9 | 6823 | -1.304 | 0.192 | 127.73 | 7417.5 | -0.224 | 0.823 | 133.80 | 6675.5 | -1.541 | 0.123 | 131.81 | 7001.5 | -0.957 | 0.339 |
| Female | 92 | 120.66 | | | | 129.88 | | | | 119.06 | | | | 122.6 | | | |
| **Age (Years)** | | | | | | | | | | | | | | | | | |
| < 37 | 119 | 114.57 | 8381 | -1.764 | 0.078 | 114.91 | 6534 | -1.652 | 0.099 | 128.26 | 6751.5 | -1.225 | 0.209 | 123.01 | 7376.5 | -0.111 | 0.912 |
| ≥ 37 | 125 | 130.05 | | | | 129.37 | | | | 117.01 | | | | 122.01 | | | |
| **Experience** | | | | | | | | | | | | | | | | | |
| <9 Years | 125 | 114.1 | 6387 | -2.371 | **0.018** | 117.56 | 6819.5 | -1.549 | 0.121 | 122.86 | 7892.5 | -0.366 | 0.714 | 118.41 | 6926 | -1.352 | 0.176 |
| ≥9 Years | 123 | 135.07 | | | | 131.56 | | | | 126.17 | | | | 130.69 | | | |
| **Patients/day** | | | | | | | | | | | | | | | | | |
| ≤ 25 | 90 | 128.46 | 7466 | -0.007 | 0.994 | 116.68 | 6406 | -1.896 | 0.058 | 157.71 | 4841 | -4.688 | **0.001** | 143.58 | 6112.5 | -2.407 | 0.016 |
| > 25 | 166 | 128.52 | | | | 134.91 | | | | 112.66 | | | | 120.32 | | | |
| **Statin prescription** | | | | | | | | | | | | | | | | | |
| ≤ 25/month | 94 | 118.2 | 6646 | -.1743 | 0.081 | 117.2 | 6551.5 | -1.875 | 0.061 | 134.88 | 7288.5 | -1.179 | 0.238 | 131.64 | 7318 | -0.519 | 0.604 |
| > 25/month | 162 | 134.48 | | | | 135.06 | | | | 123.92 | | | | 126.68 | | | |
| **Following a guideline** | | | | | | | | | | | | | | | | | |
| Yes | 137 | 145.24 | 5858 | -3.991 | **0.001** | 126.17 | 7832.5 | -0.544 | 0.586 | 138.5 | 6782 | -2.338 | 0.019 | 144.37 | 5977.5 | -3.690 | 0.001 |
| No | 119 | 109.23 | | | | 131.18 | | | | 116.99 | | | | 110.23 | | | |
| **Using a risk calculator** | | | | | | | | | | | | | | | | | |
| Yes | 84 | 150.4 | 5384 | -3.402 | **0.001** | 144.14 | 5910 | -2.381 | **0.017** | 158.58 | 4697.5 | -4.581 | **0.001** | 158.07 | 4740 | -4.478 | 0.001 |
| No | 172 | 117.8 | | | | 120.86 | | | | 113.81 | | | | 114.06 | | | |

either agree or strongly agree that ASCVD risk assessment is a vital step for the primary prevention of atherosclerotic cardiovascular diseases (93.8%), should be made an integral part of clinical practice (84.4%), and essential for initiating or deferring statin therapy (94.1%). Also, approximately 91% of participants believed that all adult patients (>40 years old) who are free of ASCVD and visiting the clinics should have a complete lipid profile for ASCVD assessment; However, physicians' positive attitude rate was less when all traditional CV risk factors were included. The high positive attitude also started to decrease when physicians were asked about their attitude towards calculating the 10-year ASCVD risk for all adult patients aged 40–75 where only 61.3% agree or strongly agree to do so. Similarly, just over three-fifths of participants (62.5%) believed that CV risk calculators as reliable tools to predict cardiovascular risk.

Comparing the attitude scores among demographic variables using the Mann-Whitney U test and the Kruskal-Wallis test are shown in Tables 4 and 5, respectively. The attitude score of participants who reported using a risk calculator was significantly higher (U = 5910, p = 0.017) than those not using one. The attitude score was not significantly associated with gender (U = 7417.5, p = 0.823), age (U = 6534, p = 0.099), experience (U = 6819.5, p = 0.121), current workplace ($\chi^2$ = 0.785, df = 2, p = 0.675), department ($\chi^2$ = 4.830, df = 3, p = 0.185), and position ($\chi^2$ = 6.869, df = 3, p = 0.076).

**Table 5. The data for the Kruskal-Wallis test for the association between demographic variables (with more than two categories) and physicians' knowledge, attitude, and practices of cardiovascular disease risk assessment.**

| Variable | N | Knowledge | | | | Attitude | | | | Practice (Risk assessment) | | | | Practice (Patient-physician counseling) | | | |
|---|---|---|---|---|---|---|---|---|---|---|---|---|---|---|---|---|---|
| | | Mean rank | Test value $(x^2)$ | Degrees of freedom | p-value | Mean rank | Test value $(x^2)$ | Degrees of freedom | p-value | Mean rank | Test value $(x^2)$ | Degrees of freedom | p-value | Mean rank | Test value $(x^2)$ | Degrees of freedom | p-value |
| **Current position** | | | | | | | | | | | | | | | | | |
| Consultant | 49 | 174.69 | 26.921 | 3 | 0.001 | 142.31 | 6.869 | 3 | 0.076 | 145.6 | 24.028 | 3 | 0.001 | 149.58 | 7.353 | 3 | 0.057 |
| Specialist | 59 | 121.31 | | | | 107.97 | | | | 147.48 | | | | 134.78 | | | |
| Resident | 50 | 126 | | | | 128.64 | | | | 145.28 | | | | 125.98 | | | |
| GP | 98 | 110.87 | | | | 133.89 | | | | 99.96 | | | | 115.46 | | | |
| **Department** | | | | | | | | | | | | | | | | | |
| Cardiology | 56 | 169.2 | 23.893 | 3 | 0.001 | 146.82 | 4.830 | 3 | 0.185 | 155.45 | 18.756 | 3 | 0.001 | 155.42 | 22.338 | 3 | 0.001 |
| Nephrology | 23 | 115.41 | | | | 118.52 | | | | 129.30 | | | | 113.96 | | | |
| Internal medicine | 110 | 121.51 | | | | 121.81 | | | | 132.62 | | | | 137.23 | | | |
| Others | 67 | 110.46 | | | | 127.59 | | | | 98.94 | | | | 96.66 | | | |
| **Current workplace** | | | | | | | | | | | | | | | | | |
| Private hospital | 82 | 125.84 | 0.165 | 2 | 0.921 | 132.46 | 0.785 | 2 | 0.675 | 123.26 | 1.044 | 2 | 0.593 | 118.88 | 5.518 | 2 | 0.063 |
| Governmental hospital | 118 | 129.74 | | | | 124.11 | | | | 133.48 | | | | 140.22 | | | |
| Private clinic | 56 | 129.79 | | | | 131.96 | | | | 125.67 | | | | 117.9 | | | |

## Practice (risk assessment for primary prevention of ASCVD)

The overall median score (IQR) for risk assessment practices was 8 (7–10) out of 15. Over half of participants (53.9%) reported either often or always recommending a lipid profile for patients aged 40–75 years for CV risk assessment purposes. Also, just around a fifth of participants (21.5%) were always screening their patients aged 40-75 years for all traditional CV risk factors, and 12.5% often did so. Moreover, a very small percentage of respondents (6.4%) reported either often or always calculating the 10-year ASCVD for their patients aged 40–75

**Table 6. Physicians' attitudes toward ASCVD risk assessment.**

| Attitudes toward ASCVD risk assessment | Agree & strongly agree (%) | Overall score Median (IQR) |
|---|---|---|
| ASCVD risk assessment is a vital step for the primary prevention of ASCVD | 240 (93.8) | 28 (27–30) |
| ASCVD risk assessment should be made an integral part of clinical practice | 216 (84.4) | |
| ASCVD Risk assessment is important for initiating or delaying statin therapy | 241 (94.1) | |
| Healthcare professionals should take the opportunity of any clinic encounter with an individual to screen for all traditional CV risks | 187 (73) | |
| All adult patients >40 years old who are free of ASCVD and visiting my clinic should have a complete lipid profile for ASCVD risk assessment | 232 (90.6) | |
| A 10-year risk calculation should be performed for all my adult patients >40 years old who are free of ASCVD | 157 (61.3) | |
| CV risk calculators are reliable tools to predict cardiovascular risk | 160 (62.5) | |

**Table 7. Physicians' practices for risk assessment and counseling before statin therapy initiation.**

| Risk assessment practices for primary prevention of ASCVD in the clinical setting | Never | Rarely | Sometimes | Often | Always | Overall score Median (IQR) |
|---|---|---|---|---|---|---|
| Screening the patients aged 40-75 years for all traditional CV risk factors | 34 (13.3) | 13 (5.1) | 122 (47.6) | 32 (12.5) | 55 (21.5) | 8 (7–10) |
| Recommending a lipid profile for the patients aged 40–75 years for CV risk assessment purposes | 8 (3.1) | 36 (14.1) | 74 (28.9) | 78 (30.5) | 60 (23.4) | |
| Calculating the 10-year ASCVD risk for the patients aged 40–75 years | 175 (68.4) | 41 (16) | 26 (10.2) | 7 (2.7) | 7 (2.7) | |
| **Patient-physician discussion and counseling practices before statin therapy initiation** | **Never** | **Rarely** | **Sometimes** | **Often** | **Always** | **Overall score Median (IQR)** |
| Discussing patient's risk for ASCVD | 22 (8.6) | 35 (13.7) | 130 (50.8) | 28 (10.9) | 41 (16) | 36 (34–39) |
| Reviewing patient's lifestyle habits ((e.g., diet, physical activity, weight or body mass index, and tobacco use) | 4 (1.6) | 19 (7.4) | 120 (46.9) | 60 (23.4) | 53 (20.7) | |
| Discussing the potential benefits of a healthy lifestyle for risk reduction | 1 (0.4) | 17 (6.6) | 43 (16.8) | 99 (38.7) | 96 (37.5) | |
| Discussing the potential benefits of statin therapy for risk reduction | 5 (2) | 6 (2.3) | 106 (41.4) | 83 (32.4) | 56 (21.9) | |
| Discussing the potential adverse effects of statin therapy | 58 (22.7) | 88 (34.4) | 50 (19.5) | 24 (9.4) | 36 (14) | |
| Explaining to the patients how and when they should take a statin medication | 0 | 4 (1.6) | 4 (1.6) | 171 (66.8) | 77 (30) | |
| Reviewing patient medications to avoid potential statin-drug interactions | 13 (5.1) | 20 (7.8) | 66 (25.8) | 26 (10.2) | 131 (51.2) | |
| Discussing the importance of adherence to a healthy lifestyle | 1 (0.4) | 4 (1.6) | 22 (8.6) | 53 (20.7) | 176 (68.7) | |
| Discussing the importance of adherence to statin therapy | 2 (0.8) | 24 (9.4) | 69 (27) | 48 (18.8) | 113 (44.1) | |
| Cost consideration (discussing the ability of the patient to pay for the medication and consider that when prescribing the anti-hyperlipidemic agent) | 29 (11.3) | 34 (13.3) | 75 (29.3) | 50 (19.5) | 68 (26.6) | |

years old. The inappropriate practices (never, rarely) were less than 20% for all three items, except for the one related to calculating the 10-year ASCVD, which was very high at 84.4% (Table 7).

Interestingly, physicians who reported seeing ≤25 patients a day had higher risk assessment practice scores than those seeing >25 patients a day (U = 4841, p = 0.001). Also, following a guideline for cholesterol management (U = 6782, p = 0.019) and using a risk calculator (U = 4697.5, p = 0.001) were associated with a higher score for risk assessment practices. Moreover, a significant difference was observed among the department ($\chi^2$ = 18.756, df = 3, p = 0.001), with a mean rank risk assessment practice score of 155.45 for cardiology, and 129.3 for nephrology, 132.62 for internal medicine, and 98.94 for others. Dunn's pairwise tests (post hoc analysis) revealed that participants from the cardiology and internal medicine departments had significantly higher practice scores than those from other departments with p values of <0.001 and 0.018, respectively. Another significant difference was noted among position groups ($\chi^2$ = 24.028, df = 3, p < 0.001), with a mean rank risk assessment practice score of 145.6 for consultants, and 147.48 for specialists, 145.28 for residents, and 99.96 for GPs. The data from post hoc analysis revealed that consultants, specialists, and residents had significantly higher practice scores than GPs.

## Practice (patient-physicians counseling before statin therapy initiation)

Regarding the counseling practices before statin therapy initiation, the physicians who participated in this survey showed suboptimal counseling practices with an overall median score (IQR) of 36 (34–39) out of 50. The highest counseling practices were educating patients when they should take statin therapy, followed by explaining the benefits of adherence to healthy lifestyles and discussing its importance for risk reduction where 96.8%, 89.4%, and 76.2% of participants reported often or always did so; respectively (Table 7). On the other hand, a low percentage of physicians (23.4%) reported either often or always educating their patients about the potential adverse effects of statin medications. Similarly, only 26.9% of participants were

often or always discussing the ASCVD risk with their patients before they were prescribed statin medication. Other counseling practice gaps were identified. In this light, only 44.1% of participants were often or always reviewing the patient's lifestyle habits ((e.g., diet, physical activity, weight or body mass index, and tobacco use) before statin therapy initiation. Moreover, just half of the respondents (51.2%) were always reviewing patients' medications to avoid potential statin-drug interactions, and a minority often did so (10.2%). Other patient-physician discussion practices are shown in Table 7.

Notably, physicians who reported seeing ≤25 patients a day had higher patient-physician counseling scores than those seeing >25 patients a day (U = 6112.5, p = 0.016). Also, following a specific guideline for cholesterol management (U = 5977.5, p = 0.001) and using a risk calculator (U = 4740, p = 0.001) were associated with a higher score for patient-physician discussion practices. Moreover, a significant difference was observed among the department ($\chi^2$ = 22.338, df = 3, p = 0.001), with a mean rank counseling score of 149.58 for cardiology, 134.78 for nephrology, 125.98 for internal medicine, and 115.46 for others. Dunn's pairwise tests revealed that participants from cardiology and internal medicine departments had significantly higher counseling practices than those from other departments with a p-value of 0.001.

## Discussion

The present study provides insights into Yemeni physicians' general awareness, knowledge, attitude, and practices toward ASCVD risk assessment. To the best of our knowledge, this is the first study in Yemen to assess physicians' knowledge and practices for ASCVD risk assessment before statin therapy initiation. Our findings show that a large proportion of physicians have not yet read the summary or full report of the 2018 ACC/AHA guideline on cholesterol management. Moreover, a significant percentage of providers were unaware of any differences between the 2013 and 2018 ACC/AHA cholesterol guidelines, were unaware of the parameters used in the Framingham and ASCVD risk calculators, and were unaware of any differences between both risk calculators. This alarmingly suboptimal general awareness about guidelines and risk calculators could lead to inappropriate practices and underutilization of statin therapy among patients with clinical indications. Indeed, our findings show that physicians who reported following a guideline performed better in knowledge and practices than those who did not.

The results also show that only around half of the physicians claimed to follow a guideline for cholesterol management in their patients. This was lower than a finding from Kuwait, in which 90% of physicians reported using a guideline [21]. Also, only a third of physicians reported using a risk calculator in clinical practice even though all physicians included are from departments that should practice risk assessment for eligible patients according to the latest guidelines recommendations [9]. This is in contrast to findings from Turkey, where authors reported more than two-thirds of physicians claimed to use risk assessment tools, and a significant proportion of them utilized guidelines for primary prevention of CVD [22].

The web version or app of the ACC/AHA Risk Calculator was developed to assist physicians in implementing shared decision-making before statin therapy initiation. However, only around 40% of participants reported being aware of the web version or the downloadable ASCVD calculator, a thirty percent lower than that reported in the USA (70.4%) [14]. The risk app not only allows for the calculation of 10-year ASCVD risk for those aged 40 to 79, but it also allows for the estimation of lifetime risk for younger adults (20–39 years old), enhancing and promoting healthy lifestyle practices early in life [23]. A lack of knowledge about this information was apparent in this study. For example, although the majority of participants (71.1%) were aware of the age category (40–79) for which a 10-year risk calculation should be

performed, a significant proportion of them was not able to identify the age category for which a lifetime risk calculation is advocated. Another knowledge gap among physicians was noted where the majority of them (82.2%) were unaware of the four outcomes captured by the 10-year ASCVD risk calculator, which was similar to one reported in the USA (85%) [14]. But it is worth mentioning that the USA study was done approximately one year after the release of the 2013 ACC/AHA guideline, while our study was done two years after the 2018 guideline release.

According to the guidelines' recommendations, if the decision about statin initiation remains unclear after estimating the 10-year risk, the physician should consider additional risk enhancers, such as the existence of chronic inflammatory conditions [9]. However, in the present study, around two-fifths of participants were not aware of this information. Low knowledge about risk estimation was reported previously to be one of the weaknesses among physicians in Saudi Arabia [24]. The identified knowledge gaps might indirectly reflect the unsatisfactory level of continuous medical education (CME) among physicians in Yemen. At the same time, it emphasizes the importance of educating physicians about the most recent and up-to-date recommendations for risk assessment and primary prevention of ASCVD.

Consultants, cardiologists, and those with higher experience years were associated with better knowledge scores. This is similar to a finding from Singapore in which authors found a higher familiarity with 2013 ACC/AHA guidelines among cardiologists than endocrinologists/ nephrologists or the GPs [16]. Also, findings from Jordan found that physicians' rank can play an essential role in clinicians' knowledge [15]. Similarly, Mosca et al. found that cardiologists and primary care physicians were substantially more aware of and incorporated CVD recommendations into practice than other specialties [25]. In contrast, McBride et al. found no substantial difference in adherence to cholesterol management recommendations between family practice physicians and internists [26].

In the USA, physicians who reported either routinely or sometimes using a risk calculator in their practice were approximately two-thirds [23]. In our analysis, those who reported the use of a risk calculator is a third (32.8), but when they were asked about their routine use in practice, only 15.6% reported a regular use (sometimes, often, or always). Eaton et al. showed a similar finding, with only 17% of family physicians reported calculating the CVD risk [27]. Also, a study from Nigeria found that only 28.4% used CVD risk assessment regularly in practice [28]. On the other hand, Alenezi et al. found that a significant proportion of family physicians (62.8%) in primary healthcare centers in Saudi Arabia reported regularly using a risk assessment tool for CVD; However, self-assessment of own knowledge was unsatisfactory among more than half of them (58.5%) [29].

Factors connected to the health system and hospitals may influence physicians' CVD assessment practices. In this context, the health system in some developed countries has integrated risk assessment tools into the patients' electronic health records. As a result, the accessibility to such tools is more convenient and the use of these risk tools has improved among healthcare providers [30]. In Yemen, on the other hand, such tools are not integrated into the health system, and patients' electronic health records are still not widely used in both the public and private sectors. Therefore, the low knowledge and suboptimal practices among Yemeni physicians could be partially interpreted by the poor accessibility to risk assessment tools during practice.

Interestingly, those who were following a guideline or using a risk calculator in their practice had higher knowledge and practices regarding ASCVD risk assessment. In a study from Australia that was done among 25 general practitioners, authors found that poor awareness of tools and guidelines was a barrier to calculating the CVD risk [31]. Also, another recent Australian study conducted among 111 general practitioners reported lower risk assessment rates

among GPs who had incorrect answers to knowledge-based questions about guidelines [32]. Another key finding is that physicians who reported seeing more patients per day had a lower level of practice, suggesting that workload could be a barrier to risk assessment and counseling practices in Yemen. Previous studies revealed that fewer time constraints would allow clinicians to use and adhere to guidelines more frequently [21, 33].

The overall attitude was high among all study participants regardless of their rank or specialty. This high positive attitude towards cardiovascular risk assessment is consistent with a finding from Nigeria. The authors found that most physicians believed in the usefulness of risk assessment in improving patient care and forming better decisions about the recommended preventive therapies [28]. Our findings of a low overall awareness and suboptimal practices combined with a highly positive attitude toward ASCVD risk assessment represent an urgent need for educational interventions. Such interventions should foster a culture of CME among physicians and strive to integrate the guidelines' risk assessment recommendations into clinical practice.

Starting a statin, which is typically a lifetime treatment, is not a simple decision, and the recent practice guidelines recommend shared decision-making between the physician and the patient before statin initiation [9, 10]. Once the patients realize their ASCVD risk, a conversation about risk-lowering interventions, such as lifestyle modifications and the use of a statin, should take place. Unfortunately, these guidelines-recommended practices were suboptimal among the participants, and many gaps were identified in the present study. In this light, a low percentage of physicians reported either often or always discussing the ASCVD risk with their patients. Discussing the patient's ASCVD risk is the first step a physician should do before statin therapy initiation as individual knowledge and perception of his own CV risk could improve the adherence to statin therapy [34].

Notably, physician-patient discussion and counseling practices regarding the potential side effects of statin therapy were also poor even though side effects have been reported to be the most common reason for statin discontinuation [35]. A previous study reported that patients' concerns and suboptimal statin usage are likely to be exacerbated by a lack of physician-patient discussion about the benefits vs. risks of statin therapy and not adequately addressing possible statin side effects [36]. Patient-physician counseling practices are essential before statin therapy initiation. These practices could strengthen the relationship between patient and physician, enhance patient engagement in this lifelong treatment decision, and improve patients' adherence to statin therapy [37].

## Study limitations

Although this study captured the knowledge, attitude, and practice of participants about risk assessment, there were some limitations. First, data collection was done only in Sana'a. This limits the generalizability of the findings. However, since Sana'a hospitals are considered referral hospitals for all governorates, it is reasonable to assume that the identified knowledge and practice gaps might be higher across the country. Second, a convenient sampling approach was utilized, and this could result in selection bias. Third, although we assessed the knowledge of risk assessment according to 2018 ACC/AHA guidelines, it is important to mention that some physicians might be adopting other guidelines. Nevertheless, our analyses found that the vast majority of those reported to follow a guideline utilized the ACC/AHA guidelines. Despite these limitations, it is the first study in Yemen to assess the risk assessment knowledge and attitude among physicians according to the latest guideline recommendations. Also, it provides valuable information about the prevalence of physician-patient discussion and counseling practices before statin therapy initiation in concordance with recommendations from the latest

clinical practice guidelines. Furthermore, the research included physicians from various departments and workplace settings with a good response rate.

## Conclusion

Physicians had overall low knowledge, suboptimal practices, and a high positive attitude toward cardiovascular risk assessment. The knowledge and practices were higher among consultants, participants from the cardiology department, those with experience years of more than nine years, and those who reported following a specific guideline for cholesterol management or using a risk calculator in their practice. However, the counseling practices were lower among physicians who reported seeing more patients per day. Therefore, physicians' training and continuing medical education regarding cholesterol and primary prevention clinical guidelines and evidence-based medicine are recommended. Also, the importance of adherence to clinical practice guidelines and their impact on clinical outcomes should be emphasized.

## Supporting information

**S1 File. Study questionnaire.**
(PDF)

**S2 File. Study dataset.**
(SAV)

## Author Contributions

**Conceptualization:** Fahmi Y. Al-Ashwal, Syed Azhar Syed Sulaiman, Siti Maisharah Sheikh Ghadzi.

**Data curation:** Fahmi Y. Al-Ashwal, Mohammed Abdullah Kubas.

**Formal analysis:** Fahmi Y. Al-Ashwal.

**Funding acquisition:** Mohammed Abdullah Kubas, Abdulsalam Halboup.

**Investigation:** Fahmi Y. Al-Ashwal, Mohammed Abdullah Kubas.

**Methodology:** Fahmi Y. Al-Ashwal, Syed Azhar Syed Sulaiman, Siti Maisharah Sheikh Ghadzi, Mohammed Abdullah Kubas, Abdulsalam Halboup.

**Resources:** Abdulsalam Halboup.

**Supervision:** Syed Azhar Syed Sulaiman, Siti Maisharah Sheikh Ghadzi, Mohammed Abdullah Kubas.

**Validation:** Syed Azhar Syed Sulaiman, Siti Maisharah Sheikh Ghadzi, Abdulsalam Halboup.

**Writing – original draft:** Fahmi Y. Al-Ashwal.

**Writing – review & editing:** Fahmi Y. Al-Ashwal, Syed Azhar Syed Sulaiman, Siti Maisharah Sheikh Ghadzi, Mohammed Abdullah Kubas, Abdulsalam Halboup.

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
