## [Decision Letter · Decision Letter 0]

11 Apr 2022

PONE-D-21-14613Risk assessment of atherosclerotic cardiovascular diseases before statin therapy initiation: Knowledge, attitude, and practice of physicians in YemenPLOS ONE

Dear Dr. Al-Ashwal,

Thank you for submitting your manuscript to PLOS ONE. After careful consideration, we feel that it has merit but does not fully meet PLOS ONE’s publication criteria as it currently stands. Therefore, we invite you to submit a revised version of the manuscript that addresses the points raised during the review process.

The manuscript has been evaluated by two reviewers, and their comments are available below.

The reviewers have raised a number of major concerns. They feel the manuscript should better frame the setting and the research question and they request improvements to the discussion aspects of the study. The reviewers also request more information regarding the study design, such as the reason for the choice of study location and participants.

Could you please carefully revise the manuscript to address all comments raised?

We look forward to receiving your revised manuscript.

Kind regards,

Jamie Royle

Staff Editor

PLOS ONE

Journal Requirements:

Reviewers' comments:

Reviewer's Responses to Questions

**Comments to the Author**

1. Is the manuscript technically sound, and do the data support the conclusions?

Reviewer #1: Partly

Reviewer #2: Yes

2. Has the statistical analysis been performed appropriately and rigorously? 

Reviewer #1: I Don't Know

Reviewer #2: Yes

3. Have the authors made all data underlying the findings in their manuscript fully available?

Reviewer #1: Yes

Reviewer #2: Yes

4. Is the manuscript presented in an intelligible fashion and written in standard English?

Reviewer #1: Yes

Reviewer #2: Yes

5. Review Comments to the Author

Reviewer #1: Thank you for this interesting Research Article “Risk assessment of atherosclerotic cardiovascular diseases before statin therapy initiation: Knowledge, attitude and practice of physicians in Yemen”

The composition of the sample is heterogeneous and the question arises whether the design has been chosen well. It remains unclear, which physician groups are regularly involved in counseling cardio-vascular diseases in Yemen. These physicians seem to be of major importance regarding the prescription of medication (statin therapy) in adherence to existing guidelines.

Additional some comments:

What is your basic population regarding the involved physicians (350 mentioned in the Abstract) or 1732 physicians (main document). What was the reason so select only an urban area?

Different physician groups are mixed (tertiary and private Hospitals and ambulant physician’s) , even physicians in vocational training.

You address the importance only of “primary prevention” in the article. What about secondary and tertiary prevention?

What was the reason to calculate a sample size for this survey?

The experts involved into the Validation seem heterogeneous. What was the rationale for their selection?

The chosen study design and the presented results of the research Article is not particularly innovative. The results are not surprising, especially since the authors themselves cite comparable results from comparable countries

Reviewer #2: Thank you for the opportunity to review this study.

The authors have conducted a cross sectional study that gave a valuable insight into physicians’ knowledge, attitudes and practices of CVD risk assessment in Yemen. The authors concluded that physicians had suboptimal knowledge and practices however they had positive attitudes towards CVD risk assessment.

I found this to be a well conducted study and an interesting read. However, there are some areas that could be improved.

Major suggestions:

Introduction section:

- I believe it would be useful for the reader to have an idea about the risk assessment tools the authors are referring to. Authors can include a brief description of the calculators in terms of factors considered in the CVD risk estimation.

Methods section:

- In sample size section, what does ‘general prescribers’ refer to?

- In the data collection section, the questionnaire components could be described in a paragraph rather than bullet points. Each section could be described as:

The first section contained ten questions regarding the participants’ age, workplace….. number of patients seen per day. The second section…

Discussion section:

- The authors state at the beginning of the discussion and then later on in the limitations section that this is the first study to assess physicians’ knowledge and attitudes. There have been a number of studies that assessed attitudes and knowledge of physicians regarding CVD assessment and prevention. Therefore, it could be better to rephrase that to first study in Yemen.

- The authors comprehensively compared the findings of their study with studies from different countries. It is important to note that countries such as US or UK have risk assessment tools that are integrated into the patient electronic health record. Therefore, the accessibility to such tools is more convenient and the familiarity and use of these tools has improved greatly among practitioners. Health system and organization-related factors could play a role in the physicians’ CVD assessment and prescribing practices. The authors could consider the accessibility and ease of use of the tools among Yemeni physicians when discussing their findings regarding physicians’ practices.

Minor suggestions:

Overall, the study is well conducted and adds to the existing knowledge about physicians’ attitudes and practices regarding CVD risk assessment. However, some sentences were phrased in an unclear way, making it difficult to follow the authors’ point. I advise the authors to work with a copyeditor to improve the flow and readability of the text.

Introduction section:

- The aims of the study at the end of the introduction could be phrased in a more concise way so that the reader understands that the authors are assessing each of the physicians’ knowledge, attitudes and practices.

Results section:

- Reporting of the number and percentages could be more consistent.

For example, the following two sentences:

the respondents were mainly general practitioners (38.3%, n=93)

Notably, only 53.5% (n= 137) of participants were following a specific guideline

Thank you

6. PLOS authors have the option to publish the peer review history of their article (what does this mean?). If published, this will include your full peer review and any attached files.

Reviewer #1: **Yes: **Christoph Heintze

Reviewer #2: No

---

## [Author Response · Author response to Decision Letter 0]

17 Apr 2022

Dear editor and reviewers,

Firstly, we would like to thank you for your precious time and the constructive comments concerning our manuscript, PONE-D-21-14613, entitled “Risk assessment of atherosclerotic cardiovascular diseases before statin therapy initiation: Knowledge, attitude, and practice of physicians in Yemen”. These comments were all valuable and helpful for improving the article. We have tried our best to modify the manuscript to meet the requirements of your respected journal. In the revised version, changes to our manuscript were highlighted in yellow as text. Point-by-point responses to the comments are listed below. The editor and reviewers’ comments are in black, and the authors’ responses are in red.

We are looking forward to your kind reply.

 

Editor comments and Journal Requirements:

Response: Thank you for your comment. In the revised manuscript, we have carefully read both files and modified the manuscript according to them. We hope the revised manuscript meets PLOS ONE's style requirements.

Response: Thank you for your comment. To clarify this, our study was part of a project that was supported by the university as an initiative without a specific grant or award number. The project was supported by providing research material only (printing out the questionnaires). In the revised submission, we have amended the funding information section to match with financial disclosure.

Response: Thank you for your comment. We have attached the data set as a supplementary file (S2 File).

 

Reviewers' comment:

Reviewer #1:

Thank you for this interesting Research Article “Risk assessment of atherosclerotic cardiovascular diseases before statin therapy initiation: Knowledge, attitude and practice of physicians in Yemen”

1. The composition of the sample is heterogeneous and the question arises whether the design has been chosen well. It remains unclear, which physician groups are regularly involved in counseling cardio-vascular diseases in Yemen. These physicians seem to be of major importance regarding the prescription of medication (statin therapy) in adherence to existing guidelines.

Response: Thank you for your comments. It is a normal circumstance for this type of study design to have different respondents to get representative physicians from all departments that manage the patients with clinical indications for statin therapy. In the methods section of our manuscript, we have determined the group of physicians who are usually involved with statin prescription and counseling cardiovascular diseases in Yemen. The groups involve doctors from the internal medicine department, cardiology, endocrinology, nephrology, and general prescribers. The statin prescription is mostly done by a physician from these departments. The general prescribers are those who are registered doctors but without any specialization, and they have the authority to prescribe statin medications. DM patients, patients with CKD, ASCVD, and primary hypercholesterolemia, for whom statin therapy could be indicated, are all managed by doctors from these departments.

We cannot for example involve physicians from cardiology only, because the knowledge may be biased and overestimated compared to those from internal medicine or endocrinology, or general prescribers. We have to include all physicians who are likely to be involved in statin prescription and CVD primary prevention regardless of rank or department. Then, we evaluate if there are any differences in knowledge between the different ranks and different departments using appropriate inferential analysis, for our data, the Kruskal-Wallis test , and the Mann-Whitney U test.

2. Additional some comments: What is your basic population regarding the involved physicians (350 mentioned in the Abstract) or 1732 physicians (main document).

Response: Thank you for your comment. To clarify this, the 1732 number in the method section is related to sample size calculation; it is the sample frame from which we selected our sample size for the study. The 350 is the number of respondents who were approached to participate in the study and received the questionnaire, and 270 of the 350 filled and handed back the questionnaire. However, 14 of the 270 questionnaires were removed due to high missing data.

In the revised manuscript, so we rephrased the sentences in the abstract:

Methods of the abstract: 

A cross-sectional study was conducted between November 2020 and January 2021. A self-administered questionnaire was distributed to 350 physicians (GPs, residents, specialists, and consultants).

Results of the abstract:

A total of 270 physicians filled the questionnaire out of 350 physicians approached, with 14 being excluded due to high missing data, giving a final response rate of 73%.

3. What was the reason so select only an urban area?

Response: Thank you for your comment. We choose Sana’a as it is the capital of Yemen, and it is the referral healthcare city for people from all governorates. Moreover, the largest private and governmental hospitals are in the capital and it is more feasible to conduct the study there. We have also acknowledged that in the limitations of the manuscript:

 (There were some limitations. First, data collection was done only in Sana’a. This limits the generalizability of the findings. However, since Sana’a hospitals are considered referral hospitals for all governorates, it is reasonable to assume that the identified knowledge and practice gaps might be higher across the country)

4. Different physician groups are mixed (tertiary and private Hospitals and ambulant physician’s) , even physicians in vocational training.

Response: Thank you for your comment. It is typical to have different physician groups for this kind of study to get a representative sample, and using the inferential analysis we can determine if there is any difference between the different groups. For example, as we have to include male and female physicians, those aged 25-40 and those older, we also have to include physicians with different ranks and from different workplaces (Governmental vs private hospitals and clinics) and departments, that prescribe statin therapy. These are demographic characteristics and it is a normal thing to be mixed, the most important thing is that he or she is prescribing or managing patients eligible for statins. If we did not include physicians with different ranks and from different departments that prescribe statins, it would be a limitation for the study and we have to report it.

5. You address the importance only of “primary prevention” in the article. What about secondary and tertiary prevention?

Response: Thank you for your comment. We addressed the importance of primary prevention because the risk assessment for CVD (our study focus) is performed for primary prevention and not for secondary prevention. According to ACC/AHA 2018 dyslipidemia guideline and the 2019 ACC/AHA primary prevention guideline, a 10-Year cardiovascular risk assessment tool is recommended for primary prevention. On the other hand, patients with preexisting cardiovascular disease (secondary prevention) already have a high or very risk for CVD and do not require a 10-Year cardiovascular risk assessment, and statin therapy is recommended for them without assessment, according to the guidelines.

6. What was the reason to calculate a sample size for this survey?

Response: We did the sample size calculation to ascertain the minimum number of participants required to detect the statistical significance. In other words, we will be able to draw conclusions with a decent amount of confidence if we use an accurate sample size calculation. Also, instead of distributing the survey to all the population of interest, we do the sample size calculation to obtain a representative sample.

7. The experts involved into the Validation seem heterogeneous. What was the rationale for their selection?

Response: Thank you for your comment. Including a multidisciplinary expert team was essential for the content validation process. Since the research involve risk assessment and statin therapy initiation and the practices regarding these two topics, this necessitates the presence of a team of experts from clinical pharmacy/pharmacy practice (statin therapy) and consultants in internal medicine (risk assessment), and expert in community medicine (demographics, sampling and CVD prevention topic). All of these experts are experienced with the guideline and research designs. These experts were asked to assess the relevance and representative of each item to its domain. Also, to provide us with feedback regarding questionnaire design, the number of items, and questions' appropriateness.

8. The chosen study design and the presented results of the research Article is not particularly innovative. The results are not surprising, especially since the authors themselves cite comparable results from comparable countries?

Response:

Thank you for your comment. The study is the first study for risk assessment among physicians in Yemen, and to the best of our knowledge, it has unique characteristics that cannot be found in any international study. In this context, all the cited studies assessed the knowledge of physicians regarding the 2013 AHA/ACC guidelines and were not specific for risk assessment. On the other hand, our study was designed to be specific for risk assessment before statin therapy initiation, and we used the 2019 and 2018 AHA/ACC guidelines. Also, most of the previously cited studies included few questions about risk assessment knowledge and were not comprehensive (knowledge, attitude, and practices of risk assessment in one study). Moreover, most of the knowledge questions are unique for this study. For example, questions number 1,2,4,5,6,7,10 (7 out of 10 ) in the knowledge section of supplementary file 1 (Section C) are unique for the present study and were extracted directly from the latest guideline recommendations. Also, physician-patient counseling practices (section F, 10 items) were unique for the present study. Yes, an item or two of these 10 practices could be found in some of the previous studies and we have to cite and compare them but all 10 items were adopted directly from the latest guideline, and no study has assessed them together.

Reviewer #2: 

Thank you for the opportunity to review this study.

The authors have conducted a cross sectional study that gave a valuable insight into physicians’ knowledge, attitudes and practices of CVD risk assessment in Yemen. The authors concluded that physicians had suboptimal knowledge and practices however they had positive attitudes towards CVD risk assessment.

I found this to be a well conducted study and an interesting read. However, there are some areas that could be improved.

1. Major suggestions:

Introduction section:

- I believe it would be useful for the reader to have an idea about the risk assessment tools the authors are referring to. Authors can include a brief description of the calculators in terms of factors considered in the CVD risk estimation.

Response: Thank you for your comments. We have added the following paragraph to the introduction of the revised manuscript:

The CV risk can be assessed using risk estimation algorithms created based on the results of cohort studies [11]. Different risk score calculators are recommended by different guidelines for assessing the 10-year cardiovascular risk [9, 12]. These risk calculators differ in the variables included and the endpoints assessed [11, 13]. For example, the 2008 Framingham General CVD risk calculator uses the variables of gender, age, total cholesterol, HDL cholesterol, systolic blood pressure, antihypertensive therapy, history of diabetes mellitus, and current smoking status [11, 13]. The outcomes being assessed are the total CVD (coronary insufficiency or angina, heart failure, Intermittent claudication, CHD death, nonfatal MI, fatal or nonfatal ischemic or hemorrhagic stroke, and transient ischemic attack). The 2013 ACC/AHA risk calculator includes almost the same parameters as the 2008 Framingham general CVD model, but in contrast to the 2008 Framingham model, it adds the race and measures only hard ASCVD endpoints (CHD death, nonfatal MI, fatal and nonfatal stroke) [11, 13].

2. Methods section:

- In sample size section, what does ‘general prescribers’ refer to?

Response: Thank you for your comments. General prescribers or general practitioners are those who finished a bachelor of medicine and were licensed without doing a specialty in any medical field.

We have changed the word to ‘general practitioners’ in the revised manuscript to be consistent, and the definition of general practitioners is included as well (licensed physicians who are graduated from an accredited medical school without being enrolled into a residency program)

3. In the data collection section, the questionnaire components could be described in a paragraph rather than bullet points. Each section could be described as:

The first section contained ten questions regarding the participants’ age, workplace….. number of patients seen per day. The second section…

Response: We appreciated your comments.

We have changed the way of describing questionnaire components according to your suggestion. The following change was made to the revised manuscript:

 The questionnaire consists of 6 sections (S1 file). Section A contained data about gender, age, working place, specialty, and experience years. Moreover, four general questions were included as follows: ‘Number of patients seen per day?’, ‘In the past month, how many times did you prescribe statin therapy?’, ‘Do you follow any clinical practice guideline for cholesterol management in your patients?’, and ‘Do you use a risk calculator for cardiovascular risk assessment in your practice?’. Section B contained 6 questions that assessed the general awareness about the 2018 ACC/AHA guideline, Framingham general CVD risk calculator, and the 10-year ASCVD risk calculator and whether they know of any differences between them. 

Section C assessed the specific knowledge regarding ASCVD risk assessment. It included 10 multiple-choice questions that were designed to assess whether physicians have the basic and necessary knowledge for risk assessment before statin therapy initiation. Section D included 7 questions in which participants were asked about their attitude towards CVD risk assessment, with responses being measured on a 5-point Likert scale: strongly disagree, disagree, neutral, agree, and strongly agree. Section E contained 3 questions that evaluated the practices for risk assessment in Yemen with five possible responses (never, rarely, sometimes, often, always). The last section (F) included 10 questions that evaluated the counseling practices of physicians before statin therapy initiation (patient-physician discussion), with responses being measured on a 5-point Likert scale: never, rarely, sometimes, often, always.

4. Discussion section:

- The authors state at the beginning of the discussion and then later on in the limitations section that this is the first study to assess physicians’ knowledge and attitudes. There have been a number of studies that assessed attitudes and knowledge of physicians regarding CVD assessment and prevention. Therefore, it could be better to rephrase that to first study in Yemen.

Response: Thank you for your comment.

We rephrased the sentence according to your comment.

5. - The authors comprehensively compared the findings of their study with studies from different countries. It is important to note that countries such as US or UK have risk assessment tools that are integrated into the patient electronic health record. Therefore, the accessibility to such tools is more convenient and the familiarity and use of these tools has improved greatly among practitioners. Health system and organization-related factors could play a role in the physicians’ CVD assessment and prescribing practices. The authors could consider the accessibility and ease of use of the tools among Yemeni physicians when discussing their findings regarding physicians’ practices.

Response: We agree with your comment, and thank you for your suggestion. The following paragraph was added to the discussion:

Factors connected to the health system and hospitals may influence physicians' CVD assessment practices. In this context, the health system in some developed countries has integrated risk assessment tools into the patients' electronic health records. As a result, the accessibility to such tools is more convenient and the use of these risk tools has improved among healthcare providers [30]. In Yemen, on the other hand, such tools are not integrated into the health system, and patients' electronic health records are still not widely used in both the public and private sectors. Therefore, the low knowledge and suboptimal practices among Yemeni physicians could be partially interpreted by the poor accessibility to risk assessment tools during practice.

6. Minor suggestions:

Overall, the study is well conducted and adds to the existing knowledge about physicians’ attitudes and practices regarding CVD risk assessment. However, some sentences were phrased in an unclear way, making it difficult to follow the authors’ point. I advise the authors to work with a copyeditor to improve the flow and readability of the text.

Response: Thank you for your suggestion. We have revised the manuscript and rephrased some of the long sentences that were highlighted in yellow. We hope in the revised version, it becomes easier for readers to follow.

7. Introduction section:

- The aims of the study at the end of the introduction could be phrased in a more concise way so that the reader understands that the authors are assessing each of the physicians’ knowledge, attitudes and practices.

Response: Thank you for your comment. 

 We replaced the old one with this sentence: 

this study aimed to evaluate the knowledge, attitude, and practices of Yemeni physicians regarding risk assessment of atherosclerotic cardiovascular diseases prior to initiating statin therapy. 

8. Results section:

- Reporting of the number and percentages could be more consistent.

For example, the following two sentences:

the respondents were mainly general practitioners (38.3%, n=93)

Notably, only 53.5% (n= 137) of participants were following a specific guideline

Response:

Thank you for your comment. We modified the results’ reporting to be more consistent according to your recommendations. We removed the numbers and kept the percentages to be consistent across all sections; since the number and percentages, both are found in the tables

---

## [Decision Letter · Decision Letter 1]

13 May 2022

Risk assessment of atherosclerotic cardiovascular diseases before statin therapy initiation: Knowledge, attitude, and practice of physicians in Yemen

PONE-D-21-14613R1

Dear Dr. Al-Ashwal,

We’re pleased to inform you that your manuscript has been judged scientifically suitable for publication and will be formally accepted for publication once it meets all outstanding technical requirements.

Kind regards,

Marianne Clemence

Staff Editor

PLOS ONE

Additional Editor Comments (optional):

Thank you for submitting your revision. After careful assessment of the revised manuscript and response to reviewers, your study has been considered suitable for publication in line with our publication criteria.

Reviewers' comments:

Reviewer's Responses to Questions

**Comments to the Author**

1. If the authors have adequately addressed your comments raised in a previous round of review and you feel that this manuscript is now acceptable for publication, you may indicate that here to bypass the “Comments to the Author” section, enter your conflict of interest statement in the “Confidential to Editor” section, and submit your "Accept" recommendation.

Reviewer #1: All comments have been addressed

2. Is the manuscript technically sound, and do the data support the conclusions?

Reviewer #1: Yes

3. Has the statistical analysis been performed appropriately and rigorously? 

Reviewer #1: Yes

4. Have the authors made all data underlying the findings in their manuscript fully available?

Reviewer #1: Yes

5. Is the manuscript presented in an intelligible fashion and written in standard English?

Reviewer #1: Yes

6. Review Comments to the Author

Reviewer #1: (No Response)

7. PLOS authors have the option to publish the peer review history of their article (what does this mean?). If published, this will include your full peer review and any attached files.

Reviewer #1: No

---

## [Editor Report · Acceptance letter]

17 May 2022

PONE-D-21-14613R1 

Risk assessment of atherosclerotic cardiovascular diseases before statin therapy initiation: Knowledge, attitude, and practice of physicians in Yemen 

Dear Dr. Al-Ashwal:

I'm pleased to inform you that your manuscript has been deemed suitable for publication in PLOS ONE. Congratulations! Your manuscript is now with our production department. 

Kind regards, 

on behalf of

Dr Marianne Clemence 

Staff Editor

PLOS ONE